# Human Herpesvirus 8 in Australia: DNAemia and Cumulative Exposure in Blood Donors

**DOI:** 10.3390/v14102185

**Published:** 2022-10-03

**Authors:** David J. Speicher, Jesse J. Fryk, Victoria Kashchuk, Helen M. Faddy, Newell W. Johnson

**Affiliations:** 1Molecular Basis of Disease Research Program, Menzies Health Institute Queensland, Griffith University, Southport, QLD 4215, Australia; 2Department of Pathobiology, Ontario Veterinary College, University of Guelph, Guelph, ON N1G 2W1, Canada; 3Novometrix Research Inc., Moffat, ON L0P 1J0, Canada; 4Natural Sciences and Mathematics, Redeemer University, Ancaster, ON L9K 1J4, Canada; 5Research and Development, Australian Red Cross Lifeblood, Kelvin Grove, QLD 4059, Australia; 6Centre for Health Analytics, Melbourne Children’s Campus, Parkville, VIC 3052, Australia; 7School of Health and Behavioral Sciences, The University of the Sunshine Coast, Petrie, QLD 4502, Australia

**Keywords:** human herpesvirus 8, Kaposi’s sarcoma-associated herpesvirus, Kaposi’s sarcoma, blood donors, transfusion transmission, safety, DNAemia

## Abstract

Human herpesvirus 8 (HHV-8), the causative agent of Kaposi’s sarcoma, multicentric Castleman’s disease and primary effusion lymphoma, predominantly manifests in immunocompromised individuals. However, infection in immunocompetent individuals does occur. The prevalence of HHV-8 exposure in blood donors from non-endemic countries ranges between 1.2% and 7.3%. Nothing was known about the prevalence in Australian blood donors. Therefore, this study investigated the active and cumulative exposure of HHV-8 in this cohort. Plasma samples (*n* = 480) were collected from eastern Australian blood donors and were tested for HHV-8 DNA by qPCR, and for HHV-8 antibodies by two different ELISAs. Samples initially positive on either ELISA were retested in duplicate on both, and on a mock-coated ELISA. Any samples positive two or three out of the three times tested on at least one ELISA, and repeat negative on the mock-coated ELISA, were assigned as repeat positive. None of the 480 samples tested contained HHV-8 DNA. Serological testing revealed 28 samples (5.83%; 95% CI: 3.74–7.93%) had antibodies to HHV-8. There was no difference (*p* > 0.05) in seropositivity between sex or with increasing age. This is the first study to show serological evidence of cumulative HHV-8 exposure and no HHV-8 DNAemia within a select blood donor population in Australia. Our molecular and serological data is consistent with published results for blood donors residing in HHV-8 non-endemic countries, which shows the prevalence to be very low.

## 1. Introduction

Human herpesvirus-8 (HHV-8), also known as Kaposi’s sarcoma-associated herpesvirus (KSHV), is the aetiological agent of Kaposi Sarcoma (KS), [1] multicentric Castleman’s disease (MCD) [2] and primary effusion lymphoma (PEL) [3]. HHV-8 is the only member of the *Rhadinovirus* genus to infect humans [4]. Infection of HHV-8 is known to be life-long, with B-cell lymphocytes, spindle and endothelial cells being the preferred targets for both its lytic and latent phases [5,6,7]. Such infection characteristics are typical of *gammaherpesviruses*. Even though HHV-8 is predominantly an opportunistic infection in human immunodeficiency virus (HIV)/acquired immunodeficiency syndrome (AIDS) patients, healthy individuals are not immune [8]. Furthermore, primary infection and reactivation of HHV-8 in these individuals can potentially be asymptomatic [4]. As HHV-8 resides in the tonsils and adenoids the primary route of infection is saliva [9,10]. However, transmission has also been described following blood transfusion from an HHV-8 positive donor to an HHV-8 negative recipient [11,12]. Therefore, there is concern that the global blood supply could be at risk of transmitting this virus, especially in endemic areas [13,14].

The seroprevalence of HHV-8 varies by geographical area, likewise the risk of transmission. Areas of the Mediterranean [15,16,17], and Central and Eastern Africa [18,19,20] are known to be HHV-8 endemic due to the high prevalence of the virus, whereas regions of Northern Europe [21,22,23,24], Southeast Asia [18,25,26,27,28], Northern [29,30] and Latin America [18,31] are non-endemic to this virus. Each of these regions have unique traits regarding how the virus is transmitted. Within non-endemic countries, HHV-8 transmission is mainly through oral-oral or oral-genital sex, due to the higher viral concentration in saliva compared to genital tract secretions [32]. This form of transmission occurs predominantly in men who engage in sex with men. Transmission through other routes has also been documented, including organ transplants and blood transfusion [4,33]. Several studies have recorded transmission of HHV-8 through kidney and liver transplants, due to leukocytes still present in the organ potentially harbouring the virus in a lytic or latent phase [34,35,36]. However, the use of leukoreduced blood may greatly reduce the risk of transmitting viruses, such as HHV-8 [32,33,37,38].

To understand the extent of exposure within blood donors and the risk of transmitting HHV-8 to a recipient, numerous blood transfusion organizations have undertaken studies examining the rate of HHV-8 DNAemia and seroprevalence of HHV-8 [22,28,29,31,39,40,41]. To date, HHV-8 DNA has been detected in two Argentinian blood donors [42]. The seroprevalence of HHV-8, on the other hand, ranges from 1.2% in Cuban blood donors [31] to 7.3% in American blood donors [29]. Currently, no studies have examined HHV-8 DNAemia, or the past exposure to HHV-8, within the Australian blood donor population. As the prevalence of HHV-8 associated disease is low in Australia, [43] it would be anticipated that the rate of DNAemia and cumulative exposure to this virus would be consistent with the literature for non-endemic countries. Therefore, to confirm this hypothesis, this study investigated the HHV-8 DNAemia, and cumulative exposure to HHV-8, within Australian blood donors.

## 2. Materials and Methods

### 2.1. Blood Donor Plasma Samples

Blood samples post-routine serology testing were collected from volunteer blood donors (*n* = 480) visiting donor centres on the Eastern side of Australia during 2013. A total of 40 samples were randomly selected for each sex in each of the age strata: 16–24, 25–34, 35–44, 45–54, 55–64, >65 years. These samples were collected in EDTA tubes (Vaccutainer^®^ Whole Blood Collection tube, Becton Dickinson and Company (BD) Biosciences, San Diego, CA, USA), centrifuged, and plasma aliquoted into 2 mL microtubes (Corning, Tewksbury, MA, USA). All plasma aliquots were then frozen at −30 °C until required. Donor demographics, consisting of age and sex, were obtained for each sample. Ethical approval was sought from the Australian Red Cross Lifeblood Human Research Ethics Committee (HFaddy150612) and the Griffith University Human Research Ethics Committee (DOH/12/09/HREC).

### 2.2. Detection of HHV-8 DNA by PCR

DNA was extracted from 200 µL plasma with the High Pure Viral Nucleic Acid Kit (Roche Diagnostics, Australia) as per the manufacturer’s protocol and eluted into 50 µL elution buffer. To monitor the efficiency and reproducibility of DNA extraction and amplification, samples were spiked with 20 µL equine herpesvirus (EHV) prior to extraction. Purified viral nucleic acid was examined for purity and concentration via NanoDrop^®^ ND-1000 Spectrophotometer (Thermofisher Scientific, Scoresby, VIC, Australia) and in triplicate for HHV-8 ORF73, ORF26, and EHV by qPCR.

Fully validated and MIQE compliant qPCR assays targeting HHV-8 ORF73 and ORF26 were previously developed by our laboratory [43]. qPCR for EHV-4 was performed using forward primer [EQHSV-330F; 5′-GATGACACTAGCGACTTCGA-3′], reverse primer [EQHSV-410R; 5′-CAGGGCAGAAACCATAGACA-3′], and probe [EQHSV-360-Pb; 5′-CTG GAGGAGGCACGCGAAA-3′] targeting an 80 bp fragment of the EHV-4 enveloped glycoprotein B [44]. In brief, qPCR was set up in 384 well plates using an epMotion^®^ (Eppendorf South Pacific Pty. Ltd., Macquarie Park, NSW, Australia). PCR master mix was made in a 10 µL volume using the LightCycler^®^ 480 SYBR Green I Master mix (Roche Diagnostics) containing 0.2 μmol/L primers (Purity: PCR/Seq; GeneWorks, Thebarton, SA, Australia), 0.1 μmol/L probe (Purity: HPLC; GeneWorks), and 2 μL of nucleic acid extract. qPCR was performed on a LightCycler^®^ 480 system (Roche Diagnostics). PCR cycling conditions consisted of a 5 min polymerase activation step at 95 °C, followed by 45 cycles of amplification (5 s at 95 °C; 30 s at 58 °C). Calibration curves were prepared by cloning amplicon into pGEM-T Easy (Promega Corporation, Auburn, VIC, Australia) and diluting 10-fold. The limit of detections of the HHV-8 ORF73 and ORF26 qPCR assays were 4.85 × 10^3^ and 5.61 × 10^2^, respectively [43]. Relative EHV-4 loads were reported in Ct values. A no template control (NTC) and a clinically HHV-8–negative control (DNA extracted from oral squamous cell carcinoma tissue) were run in duplicate in each qPCR run with the run discarded if any negative sample amplified. The standard curve was rejected if the qPCR efficiency was outside the range of 90–110%, linearity (R^2^ value) was less than 0.9800, or if any NTC amplified.

### 2.3. Detection of HHV-8 Antibodies by Serology

All plasma samples were tested for HHV-8 antibodies by an enzyme-linked immunosorbent assay (ELISA) technique developed by the Centre for Disease Control and Prevention from a published protocol [45]. The technique incorporated testing samples on two peptide-coated ELISA plates: one coated with the virion envelope protein (ORF K8.1) and the other coated with the virion capsid protein (ORF65). Briefly, ELISA plates (Microlon high-binding strip plate, Greiner Bio-One GmbH, Kremsmunster, Austria) were coated with 110 µL of ORF K8.1 or ORF65 peptides (GL Biochem Ltd., Shanghai, China) at 5 µg/mL in 0.1 M carbonate-bicarbonate solution (Life Technologies, Melbourne, VIC, Australia), or peptide-free carbonate-bicarbonate solution at 4 °C overnight. The ORF K8.1 peptide used in this study was derived from amino acid 32–62 (RSHLGFWQEG WSGQVYQDWLGRMNCSYENM T) [46], whereas ORF65 was from amino acid 140–170 (ASDILTTLSSTTETAAPAVADARKPPSGKKK) [47]. Following overnight incubation, plates were washed twice with a wash buffer (0.05% Triton X-100 (Sigma-Aldrich, St. Louis, MO, USA) in Phosphate-buffered saline (PBS)), dried and stored at −20 °C until needed. The coating of ELISA plates with the two peptides were done at the National Serological Reference Laboratory, Melbourne, Victoria, Australia, with the plates being delivered to Griffith University for testing. The coating of plates with the peptide-free carbonate-bicarbonate solution was done at Griffith University.

Plates required for testing were thawed and blocked with a blocking buffer (5% skim milk powder in PBS with 0.05% Triton X-100) for 30 min at 37 °C with gentle agitation at 50 rpm. Following two wash steps, 100 µL of samples or controls were diluted 100 times in blocking buffer, added to the wells, f and incubated for one hour at 37 °C with gentle agitation. Plates were then washed four times and incubated with a horse-radish peroxidase conjugate antibody (goat anti-human (H + L) HRP conjugate; Bio-Rad, Gladesville, NSW, Australia; diluted 1 in 8000 in blocking buffer) for 30 min at 37 °C. Following another four washes, 3, 3′, 5, 5′-Tetramethylbenzidine (TMB: Sigma-Aldrich) was added to detect bound antigen-antibody complexes and incubated for 10 min at room temperature, followed by an equal volume of 2.0 N sulphuric acid to cease the reaction. Plates were then read on a microplate reader (POLARstar Omega, BMG Labtech, Mornington, VIC, Australia) at 450 nm with a 630 nm reference and optical densities (OD) recorded.

All samples were tested on both peptide-coated plates in parallel and in duplicate. A positive result was achieved if the samples’ average OD was greater than the cut-off value of the assay. The cut-off value was calculated from the average OD of five negative controls included on the assay plus 0.150 OD units. Any sample initially positive on either or both peptide-coated plates were retested in duplicate on the ORF K8.1 and ORF65-coated plates, and in duplicate on the peptide free-coated plates to assess for non-specific reactivity. Samples that were positive on either or both peptide-coated plates two or three out of the three times tested, and repeat negative on the peptide free-coated plates, were deemed to be repeat positive.

### 2.4. Statistical Analysis

The seroprevalence of HHV-8 exposure and 95% confidence intervals were calculated and expressed as percentages. Association of previous exposure to HHV-8 with the age of the donor or their gender was examined through multivariate logistic regression analyses. The computer software Microsoft Excel (Microsoft Pty Ltd., North Ryde, NSW, Australia) and Statistical Package for the Social Sciences (SPSS: IBM Australia Ltd., St. Leonards, NSW, Australia) were used to manage and analyse the data.

## 3. Results

### 3.1. Molecular Evidence of HHV-8 DNAemia

None of the 480 samples were positive for HHV-8 ORF73 and ORF26 genes, despite all extractions and calibration curves passing quality control. EHV-4 amplified in all samples at Ct 31.7 ± 1.39 with a melt curve of 80 ± 0.227 °C confirming quality in DNA extraction and amplification.

### 3.2. Serological Evidence of HHV-8 Cumulative Exposure

Of the 480 samples tested on both the ORF K8.1 and ORF65-coated plates, 28 samples were repeat positive on at least one of the two peptides, indicating serological evidence of previous exposure to HHV-8. This equated to a rate of 5.83% (95% CI: 3.74–93) (Table 1). Interestingly, the rate of exposure to HHV-8 was equal between males and females and thus showed no association (*p* = 1.000) (Table 1 and Table 2). A trend in the decreasing rate of HHV-8 exposure with increasing age was observed from the 35–44-year age group onwards; however, there was no association between age and previous exposure (*p* = 0.76) (Table 1 and Table 2).

## 4. Discussion

Infection of HHV-8 is predominantly seen in HIV/AIDS patients with the development of KS, MCD, or PEL; however, infection still occurs in healthy individuals. Symptoms that develop during primary infection in these individuals is not distinctive and can be non-existent. Such a scenario presents an issue for safe transfusion of blood, despite the latent and lytic phases of the virus occurring within B-cell lymphocytes, spindle and endothelial cells. HHV-8 DNAemia in blood donors from non-endemic countries has only been detected in two studies from Argentina, while the cumulative exposure ranges from 1.2% to 7.3% [42,48]. This study is the first to investigate the HHV-8 DNAemia and cumulative exposure rates of HHV-8 within Australian blood donors.

Molecular testing of the 480 plasma samples showed that none had HHV-8 DNA. Such a result is consistent with the prevalence study conducted in American [29,37], and Irish blood donors [40] but different to the Argentinian blood donor study, where 2/45 (4.4%) seropositive donors had detectable HHV-8 DNA [42]. Further studies in South America showed variance (3.2–12.3%) between blood donor demographics with increased HHV-8 incidence in Native American mtDNA haplotype A2 [49].

Serological testing in this study revealed that 5.83% of samples contain antibodies to lytic HHV-8 antigens (Table 1). This finding is relatively consistent with the seroprevalence in blood donors from Argentina (7.79%) [42], Brazil (2.80%), Chile (3.00%) [41], Cuba (1.20%) [31], Hungary (2.28%) [22] and the USA (7.30%) [29], and very consistent in blood donors from China (5.70%) [28]. No association was seen in our study between the seroprevalence with sex, nor with increasing age (Table 2). These results are comparable with outcomes from Chilean blood donors [41]. For the Brazilian blood donors, only nine males were seropositive for HHV-8, indicating a strong association with males, whereas no association with increasing age was observed [41]. For Argentinian and Hungarian blood donors, there was no association with sex to previous HHV-8 exposure; however, a strong association with increasing age was observed [22,42]. These findings are quite perplexing as a definitive, universal trend does not exist. This could be related to the sample size and the socio-economical variance of the respective cohorts. Increasing the sample size and evening the distribution could accurately examine the differences in the cumulative exposure with reference to the gender and age of the blood donor as well as any association exposure with the cultural and socio-economical stance of the non-endemic countries.

A declining trend in the HHV-8 seroprevalence was seen within blood donors aged 35 years and older (Table 2); however no significant difference was observed. This result is the opposite seen in seroprevalence studies for Argentinian [42] and Hungarian [22] blood donors. This poses the question as to whether immunity to HHV-8 lytic proteins may be short-lived. Longitudinal studies examining the longevity of immunity to HHV-8 is definitive and seems to vary. Biggar et al., showed that antibodies to HHV-8 in men having sex with men were lasting up to 15 years [47]. Capobianchi et al., showed variation in immunity for a stem cell transplant patient, who died of fungal encephalitis 197 days post-transplantation [50]. Laney et al. also showed a combination of these outcomes in patients diagnosed with KS over a period of four visits [51]. The outcomes from each study represent the immunity for potentially immunocompromised individuals over time, which is not necessarily the case for immunocompetent individuals. Further studies are needed to demonstrate whether immunity to HHV-8 decreases as time progresses, or if immunity is indeed life-long, irrespective of the virus reaching its latency phase.

Our study is consistent with other serological studies in blood donors from non-endemic countries. With such a small sample size, it is not possible to assess the associated risk of transmitting HHV-8 through a blood component to a recipient. Furthermore, all platelet and red cell products manufactured within Australia undergo leukoreduction, which reduces residual leukocytes present in the final product significantly. Dollard et al. showed that in KS patients both cell-associated and cell-free HHV-8 in the blood was significantly reduced to very low or undetectable levels, depending on the pre-filtered viral load [52]. The current study supports this conclusion that the viral load within healthy individuals donating blood would be reduced to undetectable levels; however, confirmation of this speculation is required [52].

## 5. Conclusions

This study is the first to show the extent of HHV-8 exposure within the Australian blood donor population. The results herein demonstrate that no blood donors have active asymptomatic HHV-8 infection. Additionally, 5.83% of donors tested had antibodies to the virus, indicating previous exposure. Our findings are consistent with HHV-8 non-endemic countries, which also show the seroprevalence to be low or very low within an immunocompetent population.

## Figures and Tables

**Table 1 viruses-14-02185-t001:** Seroprevalence of exposure to HHV-8 in eastern Australian blood donors.

	Number of Samples (N)	Repeat Positive	
		N	% (95% CI)
Total	480	28	5.83 (3.74–7.93)
*Sex*			
Male	240	14	5.83 (2.87–8.80)
Female	240	14	5.83 (2.87–8.80)
*Age group (years)*			
16–24	80	5	6.25 (0.95–11.55)
25–34	80	2	2.50 (0.00–5.92)
35–44	80	9	11.25 (4.33–18.17)
45–54	80	8	10.00 (3.43–16.57)
55–64	80	3	3.75 (0.00–7.91)
>65	80	1	1.25 (0.00–3.68)

**Table 2 viruses-14-02185-t002:** Multivariate logistic regression analysis: influence of age or sex on cumulative exposure to HHV-8.

	Odds Ratio	95% CI	*p*-Value
*Sex (reference group: female)*			
Male	1.00	0.47–2.15	1.000
*Age group (reference group: 16–24)*			
25–34	0.39	0.07–2.04	0.262
35–44	1.90	0.61–5.95	0.269
45–54	1.67	0.52–5.33	0.389
55–64	0.58	0.14–2.53	0.473
>65	0.19	0.14–2.53	0.190

## Data Availability

Not applicable.

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
