# Peer review of "Human Herpesvirus 8 in Australia: DNAemia and Cumulative Exposure in Blood Donors"

_viruses, 2022, doi:10.3390/v14102185_

Round 1
Reviewer 1 Report
Dear Author,
The manuscript entitled “Human Herpesvirus 8 in Australia: carriage and cumulative exposure in blood donors” addresses a relevant issue in blood transfusion safety. The question underlying this study is whether or not HHV-8 virions are present in the plasma collected from Australian blood donors, and what is the seroprevalence of HHV-8 infection in this group. This could be informative regarding the risks of HHV-8 transmission by blood/blood products transfusions.
The manuscript is clearly written and I made some few remarks that I believe are important to help the authors to achieve a more clear and solid scientific report.
The first and more general observation is: why using samples from 2013 in a study submitted in 2022?
A group of more specific questions and observations:
Line 40
Rhadinovirus must be in italic since is the name of a genus.
Line 70 and throughout the text and including the title
Although understandable and acceptable in colloquial language, the term “carriage” must be replaced and fully explained. In fact, since HHV-8 is able to establish a latent infection (like all other herpesviruses), all seropositive individuals should “carry” cells harbouring the viral genomic DNA inside their nucleus. So in this context “carriage” refers to “be infected”. Yet, free-cell HHV-8 viral particles apparently are not present in the blood (according to PCR negative results from plasma samples) and this must be clearly referred.
Line 88
Please clarify the frozen temperature.
Line 94
As stated in lines 81-88 authors stored plasma not sera
Line 115
Authors should define the initials LOD.
Line 116
The superscript must be used in 103 and 102 (range of limite of detection)
Lines 133 and 134
The references 52 and 53 must be in brackets.
Line 150
Please specify the time during which the colorimetric reaction was allowed to proceed.
Line 205
For the sake of clarity, “active carriage” should be replaced by “cell-free viral particles” or conceptually-related expression.
Reviewer 2 Report
minor revision
1, page 2: The LODs of the HHV-8 ORF73 and ORF26 qPCR assays 115 were 4.85 × 103 and 5.61 × 102, respectively. 3 of 103 and 2 of 102 should be labelled as superscript
2, page 3: “The ORF K8.1 131 peptide used in this study was derived from amino acid 32 – 62 (RSHLGFWQEG 132 WSGQVYQDWL GRMNCSYENM T),52 whereas ORF65 was from amino acid 140 - 170 133 (ASDILTTLSS TTETAAPAVA DARKPPSGKK K).53 Following overnight incubation, 134 plates were washed twice with a wash buffer (0.05% Triton X-100 (Sigma-Aldrich, St 135 Louis, MO, USA) in Phosphate-buffered saline (PBS)), dried and stored at -20ËšC until 136 needed.” What do 52 and 53 here mean?
Author Response
We thank the reviewer for their comments. These were also picked up by Reviewer #1 and were addressed.
Reviewer 3 Report
The authors describe the epidemiology of HHV8 seroprevalence and virus prevalence in immunocompetent blood donors in Australia. The study adds to the body of evidence regarding the global distribution of HHV8 - an important oncogenic virus particularly in immunocompromised hosts. Although the study is descriptive and not that novel, the rationale, methods, and discussion are relevant.
Author Response
We thank the reviewer for their comments and as there were no requests for changes none have been made.